# Ecosocial Innovations and Their Capacity to Integrate Ecological, Economic and Social Sustainability Transition

**Aila-Leena Matthies [1,\*], Ingo Stamm [1], Tuuli Hirvilammi [1] and Kati Närhi [2]**

[1]   Department: Kokkola University Consortium Chydenius, University of Jyväskylä, Talonpojankatu 2b, 67700 Kokkola, Finland; ingo.stamm@chydenius.fi (I.S.); tuuli.hirvilammi@chydenius.fi (T.H.)

[2]   Department of Social Sciences and Philosophy, University of Jyväskylä, Opinkivi, Keskussairaalantie 2, 40014 Jyväskylä, Finland; kati.narhi@jyu.fi

\*   Correspondence: aila-leena.matthies@chydenius.fi

**Abstract:** The article contributes to sustainability transition research by indicating the significance of transformative grassroots innovations in the context of social work research. We introduce the integrative concept of ecosocial innovation in order to demonstrate how grassroots innovations can successfully combine social, ecological and economic aspects of a sustainability transition. By ecosocial innovations, we refer to social innovations with a strong ecological orientation (e.g., recycling workshops, urban gardening, participatory unemployment projects and new local economies). The data consists of 50 examples of ecosocial innovations in Finland, Italy, Germany, Belgium and the UK. We investigate how ecosocial innovations interconnect ecological, economic and social goals and study the factors of their integrative crucial capacity. On the basis of qualitative data analysis and thematic categorisation of ecosocial innovations, we identify five integrative practices: diversity of activities, successful networking, addressing new livelihood, focus on food and explicit conceptual work on sustainability. Very often these integrative practices emerge as pragmatic solutions to local needs. For the participants, the ecosocial innovations can be relevant sources for new livelihood and wellbeing beyond the conventional labour market. Foremost, ecosocial innovations are valuable as forerunners for sustainability transition in practice.

**Keywords:** sustainability transition research; ecosocial innovations; integration; ecological; economic and social sustainability; qualitative research; social work

---

## 1. Introduction

> *We, the world, are faced with a number of wicked problems. These wicked problems, like climate change, mobility, solitude, poverty, are complex, multidimensional and tough. They are not solved with simple solutions. We need a whole other type of thinking, developing and upscaling of new solutions. Social innovation helps to develop the methods, products, services and mindsets to create new solutions for wicked problems [1].*

The above quotation is taken from the empirical data of our research and signalises the high expectations for innovations in the context of sustainability transition research, policies and practices in addressing the complex societal challenges and the most pressing problems. In the current research on the existing social innovations, the innovations are expected to have a capacity to provide pioneering examples of and practical steps towards sustainability transition. Geels [2] describes how societal niches enable the development of radical innovations that deviate from existing regimes. Also, Gernert et al. [3] regard that the success of social innovations at the grassroots level is connected with

the societal niches where the innovations grow. In the niches, social innovations can be experimented within a context with less pressure from the mainstream society and market. They can create new pathways and pilots in inclusive and participatory processes apart from the governance of the mainstream society. Seyfang and Smith [4] demand policy and research agendas to enable the further growth of grassroots innovations regarding the societal importance of the opportunities for sustainable practices developed by them. In its recent report, The European Environmental Agency [5] gives a significant role to the potential for grassroots innovation and local initiatives to catalyse macroscale changes in the economic paradigm and cultural values, which are core drivers of unsustainability. Thus, a broad scientific consensus about the particular strengths of the position of niches to foster social innovations can be observed. At the same time, there are concerns as to whether and how their larger contribution to societal transition can be guaranteed, as discussed by Seyfang and Haxeltine [6].

While the potential and obstacles of the social innovations to transfer *across the levels* of societal sustainability transition have been analysed quite broadly, their capacity to interconnect *across the areas of ecological, economic and social sustainability*, and the corresponding disciplines and institutions, has not been investigated systematically. Regarding the complexity of the societal transition, this question is highly relevant and might also provide models for understanding the challenges of the needed systemic change beyond the grassroots level and niches. Also, O'Riordan ([7], p. 516) underlines the significance of the grassroots innovations by bearing in mind the vital need to simultaneously address planetary boundaries and rising inequalities. Avelino et al. ([8] p. 41) criticise that in the research of social innovations, the focus on human-environmental interactions is not deepened. This gap is significant since it is expected that social innovations, which aim to promote sustainability transition, will provide an elaborated model to interconnect environmental and human issues. As stated by several scholars, one of the biggest challenges for achieving sustainability is to understand better the complex interconnectivity between ecological, economic and social processes of sustainability and integrate them efficiently. According to Brandt et al. [9], transdisciplinary research on sustainability transition is still rare, although it is required for understanding the complexity of sustainability [10]. Fischer-Kowalski and Rothmans [11] argue that the persistent challenges of current society require the transition of structure, culture and practices of societal systems. Also, Loorbach et al. [12] regard it as a core ambition of the sustainability transition research to understand that the major societal challenges can be countered only through fundamental systemic change across the ecosystems, economy and societal regimes. Efforts to foster integrative transdisciplinary thinking are mainly emerging as theoretical models of macro-level systems or at the paradigmatic-conceptual level of sciences [13,14]. The integration of the areas of sustainability in the practice of social innovations is thus worth taking a closer look at.

The aim of our article is to contribute to the understanding of the complex interconnectivity between ecological, economic and social sustainability transition by investigating how the social innovations expressed by grassroots initiatives integrate these sustainability areas. In the context of our qualitative empirical research, we have launched the concept of *ecosocial innovations* (ESI), whereby we refer to grassroots level social innovations that combine ecological and social goal setting. This paper, which is based on a four-year research project, gives an overview of such ESIs in five European countries and analyses their integrative capacity regarding the ecological, economic and social fields of a sustainability transition. We first describe what kind of ESIs as niches of transition [2,4] can be found in Finland, Germany, Belgium, Italy and the UK. Secondly, we analyse, applying the thinking of Raworth [15,16], how the ESIs apply to sustainability transition where the planetary boundaries, the social foundation and the human needs, as well as the search for a new type of regenerative economy, are all present and interconnected at the same time in the concrete practices of ESIs. Finally, we interpret in more detail which factors may explain this particular integrative capacity of the ESIs. Whether these integrative capacities could scale up as models for more systemic and holistic thinking in sustainability transition of society at large is discussed shortly at the end of the paper.

## 2. Theory and Concepts

The concept of ecosocial innovation (ESI) used in this paper needs clarification as the research area of social innovations is already quite rich with conceptual approaches, as Lubberink et al. [17] recently stated. The concept of ecosocial innovation is rooted in the ecosocial paradigm, which has developed mainly in the European context of social work [18,19] and social policy [20,21] since the 1980s and has connected these fields with the ecological movement, policies and research. It frames the research tradition and disciplinary context also in the research project behind this paper. Our research's premise is that ESIs have become relevant for social work due to their potential capacity to meet the needs of local communities in an ecologically sustainable way, in particular, by offering opportunities of social inclusion for people in precarious situations and by searching for a livelihood beyond the mainstream labour market (see O'Riordan [22]). To pave the way for a more sustainable social work, the research project elaborates on the potential of ESIs and discusses their role in sustainability transition and in social work. However, in this paper, we are addressing only the way how ESIs bring the various fields of sustainability together in their practices.

The theoretical premises of this article are based on the ecosocial paradigm and the role of social innovations in sustainability transition. The rather optimistic view on social innovations is crystallised in the International Handbook of Social Innovations, where Parra [23] places social innovations in very close connection to social sustainability. She defines social sustainability in terms of equity and justice and examines the 'social' by signifying the models in which human beings live together, build societies interactively and search for alternatives to address ecological challenges. For her, 'social innovation consists of satisfaction of human needs, changes in social relationships and increasing socio-political capability' (p. 147). She argues that social innovations have the capacity to enforce social sustainability especially through innovations, such as socially sustainable 'greener' lifestyles, projects reconnecting human beings with nature and research for building sustainability with participatory action research (also Borström [24]). While reflecting the chances of effective realisation of the UN Sustainable Development Goals, Hajer et al. [25] also highlight the importance of grassroots innovations. Sustainability transition needs the involvement of new agencies beyond the top-down governmental efforts that the authors call 'cockpitnism'. The authors firmly believe in the civil society and the local level as part of an 'energetic society' that can help make the sustainability goals become a reality and in 'green competition that can initiate novel ideas and technologies and stimulate new business practices' (p. 1657). Most of the optimistic interpretations of the role of social innovations can be framed theoretically by the broadly used multi-level perspective theory of transition (MLP) (see Geels [2]), which emphasises the role of local social innovations created in niches, to the large-scale system changes at the levels of regimes and landscapes [2]. Instead of incremental innovations as gradual improvements in existing technologies and practices, radical innovations provide new grassroots models, which are greatly challenging the mainstream thinking in the field concerned [26].

Regarding the conceptual debate, Haxeltine et al. [27] have identified three categories of innovations when analysing the concept of social innovations from a sustainability transitions perspective as part of a systemic change: grassroots social innovations responding to social demands, broader initiatives addressing society as a whole and systemic type initiatives influencing fundamental changes in values and policies reshaping society. Seyfang and Smith [4] analysed, in the UK context, two different strands to innovations in sustainability transition: these are ecological modernisation and technological innovation on the one hand, and the community action and the social economy on the other hand. The authors suggest overcoming this dichotomy with the concept of grassroots innovations. In a similar way, Geels [2] underlines the distinction between system level technological and social innovations on the one hand, and 'radical innovations' in the niches of the grassroots level on the other hand. Seyfang and Smith [4] have argued that the grassroots level is a neglected site of innovation research for sustainable development, while macro-level technological innovations have a prominent place in understanding sustainability transition through innovations. Seyfang and Haxeltine [6] underline the significance of civil-society-based social innovations for the sustainability

transition. The authors also discuss the limitations of grassroots innovations and apply the strategic niche management (SNM) theory to identify and overcome the critical factors. According to this theory, there are three key processes which may hinder or enable the innovations developed in the niches at the grassroots level to be upscaled and achieve an impact at the regime- and landscape level. The expectations from inside and outside need to be realistic (1). Networking also outside of the niches with resourceful stakeholders is essential (2). Finally, social and experiential learning strategies (3) in community-based activities through doing things is more effective for change than educational information-giving.

In our conceptual thinking of ESIs, we follow the idea of Halbe et al. [28] where the usage of models in transition research builds upon real existing practices and not upon simulated models of transition. The approach of social-ecological systems emphasises how communities, societies, economies and cultures are all embedded in the biosphere and how the interaction between people and the biosphere goes in two directions: social systems are shaping the biosphere, but they are also shaped by and dependent on the biosphere [29]. Although the studies of Raworth [15,16,30] mainly address the need for new economic thinking, her way of formulating the immanent interdependency between ecological, economic and social development provides a plausible frame for our discussion, too. She combines the nine planetary boundaries identified by Rockström et al. ([31]; see also, e.g., Steffen et al. [32]) with the elements of social foundation and illustrates them with a doughnut figure that provides a frame to encompass human wellbeing and redesign economies [24,32]. We regard Raworth's formulation as both a normative agenda and as a useful analytical frame to be applied for investigating how sustainability transition is practised in regard to the integration of the ecological ceiling and the social foundation. From this perspective, our assumption is that ESIs are taking steps towards the regenerative economy that could function within the safe and just space of humanity. Further, her detailed description [15] of the social wellbeing of human communities can be applied for analysing the sustainability-relevant value of social innovations. The ESIs may reflect the doughnut model if they limit their activity in the 'doughnut' between the social foundation and the ecological ceiling. This means that they satisfy social needs of communities and individuals without overshooting the ecological limits in the use of natural resources and develop a non-profit economy serving human needs and supporting the regeneration of the environment [15,16].

Our research is inspired by the transformative potential that these new societal solutions beyond the mainstream economy may have in relation to the major societal challenge of social inclusion and the need for a fair share of wellbeing [22,33]. At the same time, we argue that the ESIs demonstrate in practice how to overcome the thinking about sustainability transition in separated silos of the environment, the economy and social wellbeing ([34], p. 47).

We regard ESIs as part of the civil-society-based social innovations and grassroots level [4]. Concerning the societal position of ESIs, our conceptual interpretation of them is similar to the understanding of social innovation as grassroots initiatives ([3], pp. 21–22) or radical innovations [5]. However, in contrast to emerging research on innovations [3,4], we do not follow up on the comprehensive processes or study the impacts of the transformative capacity of the ESIs in this paper. Instead, we focus on the question of how the ESIs are able to establish an integrative connection between ecological, economic and social sustainability in their activities.

## 3. Material and Methods

The initial mapping process of the field in our research project aimed to give an overview of what kind of ESIs exist in Finland, Germany, Belgium, Italy and the UK. This was done in order to be able to choose the most relevant examples for our later case studies (see Stamm et al. [35]). For this purpose, we defined 'an ideal type of ESI' as a tool for our empirical research on behalf of the following three criteria [36]. We included in our data on ESIs such grassroots activities which could at least at some level fulfil these three criteria: (1) they are developing new innovative practical steps towards a more sustainable society and are part of a social or solidarity economy and are not (or not only) aiming

to make profit (see Longhurst et al. [36]); (2) they enable the participation and realisation of the new ideas of people in unemployment and are, in particular, working for and with young people; and (3) their activities include ecological sustainability in one or another way, for instance, by enhancing fair distribution of material resources and reducing environmental impacts in their own activities and in the communities in which they are enrooted.

The ESIs we found are local organisations, initiatives or associations that are tackling ecological and social challenges in the field of social and solidarity economy with innovative solutions. Besides Finland where the research is affiliated, a further four European countries, Belgium, Germany, Italy and the UK, were selected due to our pre-knowledge about their innovative projects and our contacts with researchers, who could provide further information and contacts regarding this search. Also, a diversity of European types of welfare state contexts were included with this selection (according to Esping-Andersen [37]). In Finland, the mapping phase was conducted by posting a call for ESIs with the criteria in existing relevant e-mail lists and in social media groups of social work and various associations, as well as with the help of already existing academic and professional contacts. In Belgium (Leuven), Italy (Bolzano) and the UK (Durham), our first contact points were academic experts in the field of social innovation, social economy and sustainability studies, as well as the region around these universities. In Germany, in addition to existing contacts with researchers in Berlin, the search was able to build upon the personal knowledge and contacts of a member in the research team due to his previous work in Berlin.

The aim of the mapping phase was not to achieve a comprehensive collection of all existing ESIs, but rather the aim was to achieve a general picture of the various types of projects, initiatives and activities. In this phase, when we focused on the general information about the variety of the ESIs, the data collected was not so exact as in the latter detailed case studies [20]. However, while selecting the cases, we already realised that the mapping data actually allows an exciting analysis on the general tendencies of ESIs to consider the question of how ecological, economic and social sustainability are interconnected in the practices and the role which the innovations can play in sustainability transition.

Applying the ideal type of ESIs as a criterion, we documented altogether 50 examples of ESIs in our data: 22 from Finland, 9 from Germany, 7 from the UK, 7 from Italy and 5 from Belgium. The ESIs included in our data are listed in Appendix A. Our research team contacted most of them personally, by phone, email or a visit, and provided a description of them according to the three descriptive criteria. In Finland, we searched for every ESI across the country, whereas in the other countries, the search targeted only a certain region or city where we had contacts and direct access to the research field. The aim was not to achieve a systematic collection, but rather it was to reflect the variety of the existing types of projects, initiatives and activities.

The methodological approach we used in analysing the qualitative data for this paper follows an interpretative phenomenological approach [38] and aims to understand the significance of ESIs as a phenomenon in the context of sustainability transition research and to provide descriptive knowledge. We applied a thematic analysis [39,40] approach to the information texts of the ESIs, as well as to the research notes and interviews of ESIs, we visited or with whom we had a telephone or online interviews and email correspondence. We first conducted a descriptive coding of textual data about the content of the ESIs according to the three criteria of the data collection. We listed each of the 50 ESIs in a table and described how they correspond with each of the three criteria. This categorisation addressed the second research objective and provided systematised information on how the contents of ESIs mirrored the awareness of the ecological borders, the social needs of people and the search for an alternative economy. In the second round of the analysis, we coded textual references, which allowed us to interpret the way how the ESIs integrate the different areas of sustainability and which factors enable this integrative capacity. Finally, we categorised five integrative factors from these references.

## 4. Results

### 4.1. Overview and Types of the Mapped Ecosocial Innovations

The data analysis provides a descriptive view of the 50 ESIs regarding their activities and development. It gives an overview of a broad diversity of the ESIs, including re- and upcycling projects, gardening and agricultural projects and projects avoiding food waste, social enterprises, cooperatives and community economy models, employment projects and spaces for creative and social community building. As grassroots initiatives, most projects typically created fair and low-threshold jobs based on creative upcycling of waste materials or providing services. The large emergence of a variety of organisations based on the social and solidarity economy and cooperatives at the local level can be seen as combining social goals with economic activities and broadening the dominating narrow understanding of the economy [41]. Further, diverse types of spaces emerged for community building, such as meeting points, neighbourhood and cultural cafés, and restaurants against food waste. In many cases, they were connected to urban gardening or recycling shops and handicrafts. We have mainly focused on the project types of innovations and have not looked at innovations related to lifestyles and participatory local communities as such. Some of the ESIs are nationally networked like the Finnish National Workshop Association (NWA), which brings together around 250 local organisations that offer training and employment for people at the margins of the labour market. Their local practical workshops are mainly based on recycling, upcycling and service provision [42]. Others may be part of international networks, like the Durham REfUSE Initiative [43], which is part of the global Real-Jung-Food network. At the local level, it runs pop-up restaurant events and community campaigns against food waste.

Some of the ESIs are forerunners of alternative projects that were developed in the 1980s and are now established, while others may have started very recently. Based on the different narratives that the ESIs use to describe themselves at the general level and their gradual development in merging the various areas of sustainability transition, we can categorise four types of ecosocial innovations. First, there are associations and projects *starting from a social purpose and interest*, in particular, to mitigate unemployment, youth unemployment or create alternative ways of working in practices based on recycling or upcycling; for instance, youth workshops, which are often combined with socio-cultural activities. Most of the data collected from Finland address these kinds of challenges and new practices; for instance, Valtaajat (the Claimants), Metropolitan Area Reuse Centre, Kokkotyö (Kokko-Work in Kokkola), After Eight in Pietarsaari and Uusiotuote (Retro product) in Jyväskylä. Also, the Life e.V. from Berlin and VELO from Leuven in Belgium, as well as Albatross, Clab and Akrabat from Bolzano in Italy and Re-f-Use in Durham, represent ecosocial innovations that have their roots in social goals.

Second, there are ESIs that have *started from strong environmental awareness* that aims at ecologically sustainable wellbeing and production, for example, food co-ops and urban gardening, organic agriculture, Green Care- projects or outdoor activities as social rehabilitation. Such examples from Finland include Oma Maa (Own Land) in Helsinki, Luontopolku (Nature Path) in Tampere and Luontopaja (Nature Workshop) in Pori. Further, we found Kunst-Stoffe and Klima-Werkstatt (Climate-Workshop), Prinzessingarten and Real-Jungk-Food in Berlin, Vinterra in Italy and Fruitful Durham in the UK.

The third type of ESIs is mainly ESIs with a priority on *culture, education and community building* with a variety of meeting points, cultural cafés and training offers. Examples from Finland are Lapinlahden Lähde in Helsinki (Spring of Lapinlahti suburb), Hirvitalo-House (Elg House) in Tampere and Bike Workshop in Kotka. Further, we document similar examples, such as Haus der Eigenarbeit (House of Own Work) in Berlin and Repair Café and Riso community centre in Leuven.

The fourth type we documented is *hubs, umbrella organisations or houses and research networks of ESIs*; for instance, the Social Innovation Factory in Belgium, Project Haus in Potsdam, Germany and Ideas Hive in Durham. Also, the local forms of the Transition Movement can be counted in this typology.

Altogether, 23 ESIs correspond best with the first (social) type, 16 with the second (ecological) type and seven with the (cultural-community) third type. Actually, three ESIs included umbrella functions of the fourth type, but this type overlaps greatly with the others. Further, the typologisation is not very clear cut, since different participants involved in the ESIs may have different priorities which can shift during the lifespan of the ESI. At the grassroots levels, ecological steps of transition reinforce pressure towards environmental sustainability with the focus on social components, for example, community building, training and work possibilities.

*4.2. The Content of the ESIs Regarding Ecological Ceiling, Social Needs and Transition of Economy*

Our second research question aims at a more detailed reflection of the core processes of sustainability transition applied in the ESIs [41]. The references about contribution to *ecological sustainability transition* in our data demonstrate that part of the ESIs had a clear focus on ecological issues, for instance, in developing practical actions on mobility by bikes, like VELO in Leuven, and improving the living environment with urban gardening. However, most frequent were references to recycling and upcycling projects, which have the ecological impact of a reduction in the use of natural resources. In many cases, this application of the circular economy built the material economic base for the ESI, too. Re-usage of materials also enabled a base for job creation for the participants as the satisfaction of core social needs. This is visible, for example, in many Finnish recycling workshops and also aims to integrate unemployed people into the labour market. Another typical field of ESIs is projects against food waste and developing social restaurants and urban gardening, like Re-f-Use project in Durham. Environmental sustainability is promoted by re-using the built urban environment and empty or waste spaces. Re-use of the rural environment appears in projects for sustainable agriculture and eco-tourism, which was typical, in particular, in the region of South-Tyrol in Italy. Many ESIs were part of the Agenda 21 or Transition Movement, too, like Transition Durham. The ecological principles may also strongly dominate in individual decision making, which is seen in the following quotation:

> *I come also from social work, but I wanted to do something really meaningful, so an ecological approach was important to me, because that is important for my whole life.*

> *(Interviewee, Kunst-Stoffe upcycling project, Berlin)*

We considered that contributions to transition *towards social sustainability transition* by the ESIs are demonstrated in their aims to meet the social needs of participants and further target groups and to safeguard some of the social foundations of human communities [16]. The majority of such references are about practical offers or job and income, strengthening communities or developing new understandings of wellbeing [44]. This was typically expressed as follows:

> *Wertraum is open for people who are long-term unemployed, who are facing 'multiple obstacles' when searching for paid work, also young people, but also former refugees, people with disabilities and so forth; it has an inclusive approach.*

> *(Field research notes, Die Wille, a social enterprise, Berlin)*

Furthermore, we discovered references to the development of sustainable local communities by means which included food production and avoiding food waste. Social needs were visible also in the establishment of communicative socio-cultural infrastructures—such as open community meeting points, cafés, cultural spaces and events, like Project-House in Berlin or library-Cafes and Hirvitalo in Finland. In many ESIs, the interests of marginalised people in vulnerable situations were addressed, in particular, regarding their employment and training possibilities. In comparison with stressful experiences in the mainstream labour market, alternative forms of work or other meaningful and sustainable activities were developed to promote wellbeing. However, in many cases, the aim was not just to have another type of job. This need was immediately interlinked with the development of alternative forms of economy, the circular economy with recycling, for instance. In doing so, meeting

the social needs of participants and developing new forms of subsistence were integrated into both the ecological and economic sustainability transition.

Kemp et al. ([45], p.79) talk about the 'humanisation of the economy' while analysing the way transformative social innovations challenging the marketised and bureaucratised mainstream economy with the unconventional and non-hierarchical organisational culture. In our research, the data referring to *economic transition* demonstrates an intensive search for alternative forms of economy. In particular, recycling and upcycling build a base for a new economy, which takes a form of the circular economy, solidarity economy and exchange economy [42]. This search is very concrete, as seen in the quotation:

> *It is a cooperative—a new form of community cooperative. N.N. is also telling me, that currently there is a lot of discussion about should there be a national law on community cooperatives or should the practical experience first show what works.*
>
> *(Field research notes, Mals Cooperative, Italy)*

### 4.3. The Factors Enabling the Integration of the Various Fields of Sustainability Transition

Our third research question aims to interpret the factors which explain why the ESIs are able to integrate ecological, economic and social sustainability transition in their practice. We asked which particular characteristics of their activities allow this crucial capability. We identified five such integrative factors: diversity of activities, networking, addressing new livelihood, focus on food and explicit conceptual work on sustainability.

*The diversity of activities* is one of the unique joint features in most of the ESIs. All ESIs combine various activities in their everyday life without separating the ecological, social, cultural and economic spheres. The process of combining new forms of actions takes place flexibly based on emerging needs and gives space to new participants with new ideas. The following quotations from the data describe this feature:

> There are workspaces for textiles, wood, pottery, ( . . . ) a repair café, a bike repair place, an oven for baking bread, a learning space for educational projects, an office for counselling and supporting other projects to get started or develop ( . . . ) Refugee organisations have their offices in our buildings - Some gardening is done ( . . . ) a basic income project.
>
> (Field research notes, Projekthaus Potsdam, community project house, Potsdam)

However, due to this mixture they also face obstacles in the current bureaucratic-technical structures:

> Sometimes when they apply for money people say that they are at the wrong place because it is somewhere in-between the social, ecological, economic fields.
>
> (Kunst-stoffe, an artistic upcycling project of materials, Berlin, field research notes)

A further typical feature of *networking* with other similar or complementary groups explains, obviously, how the ESIs succeed in interconnecting the various areas of sustainability transitions. Many of the ESIs report about practical and ideological external collaboration in extended networks with a variety of transformative organisations. The ideological networking can take place even internationally, like in the Transition Network. More practical networking means a joint regional or local hub for sharing information and for improving visibility, as demonstrated in the following:

> Ideas Hive is a project which creates spaces for communities ( . . . ) to bring people together from across local communities in Co Durham. We host conversations about ( . . . ) how we can improve our areas ( . . . .) being joined with a funding body so if any great ideas come out of the evenings we can also help you to get them off the ground.
>
> (webpage, Ideas have, regional—the hub of transformative groups, Durham)

Both the diversity of actions and networking describe pragmatic and functional ways of working and creating solutions in the ESIs. Adding new activities to the growing content of an ESI develops gradually during the life circle of the ESI in the faced situations rather than through a conscious aim from the start. This pragmatic approach of the ESIs gets enriched by the involvement of local citizens and practitioners as a core strength of the integrative approach as well ([6], p. 7). The ESIs' pragmatic interconnectivity emerges from the various needs, interests and competencies of the participants, and it appears as an important option for sharing knowledge, resources and ideas. However, combining diverse activities and networking with others does not only enable mutual support but it obviously also deepens the understanding of their mutual interconnectivity. With their pragmatic way of integration, the ESIs actually demonstrate a merger of ecological, economic and social areas at a local micro-level that is required by the sustainability management research at the macro-level of ecological and economic systems [14].

Further, under the theme of *'New livelihood'*, we categorised the broad and intensive thematic content of ESIs by referring to the creation of substituted jobs, new employment, labour-related training and sources of income. The core aim of most of them is to enable people's participation in meaningful activities and to work in a community on an ecologically sustainable basis. Often, this means a direct connection to the social and solidarity economy, like a cooperative. Most excitingly, there emerges a combination of all these goals:

> It is a quite new social cooperative, 2 years old. The main idea was to combine organic farming while supporting marginalised people, mainly people with mental illnesses.
>
> > (field research notes, Vinterra, Italy)

Some of the ESIs are essentially based on the use of unpaid work, which enables community participation, in particular, in the UK:

> We have enough volunteers, more than we can take.
>
> > (Interview, Re-f-Use, against food waste—project, Durham)

Also, *the focus on food* builds another theme addressed in the ESIs' activities, which makes it self-evident that the borders between ecological, economic and social transition are not only crossed over but are also disappearing. This deep interlinkage is demonstrated, for instance, in the researcher's notes from the site visit to such a project in Helsinki:

> Transformative: towards solidarity economy building and sustainable food culture;
>
> - alternative: developing new ways to participate in food production;
>
> - co-operative as a model and doing a lot of voluntary work (visit notes, Oma Maa, organic food cooperative and farm).

Martiskainen [46] argues that the new technological and social innovations have emerged to deal with society's problems, especially regarding the essential systems connected with the everyday life of communities (also Seyfang [47]). This is also the case in the ESIs that address livelihood and food as increasingly critical issues of the current societies. The interlinkage between these core social foundations of human life and the environment and the economy is immanent and plausible as systems-independencies [48] not only at the everyday level but also at the political macro level. By searching for new types of work beyond the current labour market, the ESIs demonstrate a critical change in the meaning of work, social inclusion and income. If food is regarded as a common of a community instead of a commodity, the immanent linkage from food to the economic and ecological spheres becomes visible, and so does the linkage to transformative power [49]. Through these features, the ESIs become part of larger movements that move towards the transition of work and food policies and, thus, include the potential to scale up to the global level [50].

Finally, we could identify the features from the data to *explicitly address sustainable development* and work conceptually on it in the activities of the ESIs, which self-evidently interlinks the various

fields of sustainability in an integrative way. Such references in our data paint a picture of a variety of conceptual work and raising awareness about sustainability transition through open training offers, workshops, targeted courses, events, research, single actions and programmes. As an example, Transition Durham has established a Research Group (interview notes, Transition Durham), which

> ... carries out and encourages research which can contribute to the projects and overall aims of Transition Durham. Currently, the main projects of the Research group are related to creating a research directory and creating an alternative space for higher education teaching and research.

> (Webpage of Transition Durham)

This feature demonstrates the cognitive reflection and self-understanding of the ESIs. The way such ESIs discuss sustainability is inter-sectoral or directly embedded in a comprehensive frame—like in the Transition Movement and Transition Cities or Agenda21 [51]. Such qualities are characteristic for 'radical innovations', that is, alternatives to the mainstream in the particular field [5]. The conceptual activities are also found in the networks, which maintain, for instance, digital knowledge bases, criteria debates and awards, for example, the Belgian Social Innovation Factory. As such, the conceptualisation of sustainability in its integrated form makes the embedded values of the ESIs visible. It enforces them to encounter the mainstream. However, most of the ESIs do not address such conceptual reflection but can rather be described as 'quiet sustainability' [52]. It can enforce sustainability with traditional practices, that are invisible, informal actions and other activities that are not explicitly recognised as sustainability transition.

## 5. Discussion

In this article, we provide strong evidence that more than a thin network of innovative experiences of sustainability transition exist in European countries at the grassroots level of local communities. Even though we are aware of the limitations of the data, which is based on a quite general mapping of 50 ESIs in selected geographic spaces from a particular perspective of social work research, it is obvious that a remarkable number of such innovative projects can be found in many regions in Europe.

The relevant existing research of social innovations in the sustainability transition research, such as by Geels and Schot [2], Gernert et al. [3], Seyfang and Smith [4], as well as Seyfang and Haxeltine [6], has provided valuable results regarding the promising role of the innovations from different perspectives. They mostly see the promising role of social innovation as being connected to the benefits of their particular position in the societal niches. At the same time, most authors realise that the innovations may face challenges when aiming to transfer vertically upwards and to get scaled up across the societies. Solutions to these challenges are developed, too (see Seyfang and Haxeltine [6]). The novelty of our paper is that we have analysed the capacity of the ESIs to interconnect horizontally across the areas of an ecological, economic and social sustainability transition. This capacity is one of the most significant ones as it offers a fundamental pathway to approach the challenging complexity of sustainability transition and the systemic character of the necessary societal changes. We can consider that actually all of the documented 50 examples of ESIs interlink the three areas to some extent since this was also a criterion for selecting them in the data. However, there are variations between the ESIs in the priority given especially to ecological or to social sustainability areas. These variations enabled a slight typologisation of the ESIs. Finally, we interpreted five characteristics of the ESIs which explain the integrative capacity of the ESIs. These are: diversity of activities, networking, addressing new livelihood, focus on food and explicit conceptual work on sustainability.

Since our research questions differ essentially from the previous social innovation research, a direct comparison is not meaningful. We can, nevertheless, combine the results and discuss them jointly. For instance, it is to be assumed that the five factors explaining the integrative capacity of the ESIs are in a similar way also basically connected with the niche-position of the ESIs, and what has been said in previous research about the benefits of niche-innovations applies also for the ESIs.

Therefore, ESIs may also share the same limitations regarding the enlargement of innovations to further levels of a sustainability transition. Interesting parallels can be identified between our results and those by Seyfang and Smith [4] who analysed the characteristics of grassroots innovations. While we found that the diversity of activities inside ESIs is a strong factor that enables to interconnect ecological, economic and social sustainability, Seyfang and Smith (ibid.) also see diffusion as a benefit in the activities of grassroots innovations, for instance, in Time Bank. However, they also regard the diffusion as a challenge for internal cohesion and external operations, especially in regard to the policy interventions. Surely, similar phenomena may be found in ESIs in a deeper investigation. Further, Seyfang and Haxeltine [6] also identify the value of networking for the upscaling of grassroots innovations. However, they emphasise networking with resourceful external stakeholders, while we realise the importance of networking with other ESIs in the regions to enable the integration of the various areas of sustainability. We argue that each of the five integrative factors of the ESIs can be upscaled and apply at the level of regimes as systemic models, too. They may be promoted by landscape factors like changes in the labour market, consumer behaviour and technological innovations. Recycling and urban gardening are already 'trendy'.

The transformative potential of the niches of ESIs can be discussed in the light of the doughnuts model of Raworth [15,16]. The most promising innovative contribution of the ESIs is that they present concrete practices of ways to satisfy human needs and wellbeing that aim to avoid further overshooting of the ecological ceiling and strengthen such economic models that serve the community instead of economic growth, as is suggested in the doughnuts model [16]. Due to their integrative capacity, the ESIs can deliver integrated solutions for several dimensions of social foundation at the same time, such as food, income and work, education, social equity and health. Such holistic approaches are important for achieving sustainable wellbeing [53]. Therefore, the upscaling of ESIs as an integrative model of wellbeing beyond the economic growth-dependency and destructive work-orientation of the current society would contribute essentially to the sustainability transition. At the same time, the activities, services and production processes that are developed in ESIs provide solutions for more sustainable use of natural resources and means to reduce carbon emissions.

## 6. Conclusions

The policy directions to be proposed suggest that it is obvious that the upscaling of the socially and ecologically valuable innovative activities as piloted in the ESIs require changes in the economic systems, including social security and labour policies. The data analysed for this paper does not allow to investigate deeper the economic base of the ESIs, which is addressed in forthcoming papers based on the case studies of ESIs. Nevertheless, the models of community economies [37], social and solidarity economy [54] and the circular economy [55], including critical reflection of their limits, are essential directions to allow the ESIs to grow. For instance, Falcone et al. [56] identify a landscape level of pressure in Italy—in accordance with the MLP theory [2] to a transition towards a greener economy and investments in sustainable innovations. At the same time, the systems of income security and labour market, especially the activation programmes, can enable or restrict the financial possibilities to participate in ESIs. The European welfare states are increasingly struggling between the need for social inclusion of people distanced from the labour market and the pressure to reduce the costs in social investments. This discrepancy reflects the two different types of understanding of active social citizenship as recently conceptualised by Eggers et al. [57]. In the *self-determination type*, the state offers support for social security and services, which enable self-determined active social citizenship 'in terms of choice and autonomy' ([57] p. 48). In the *self-reliance type*, the state forces citizens to be self-reliant and finance and organise their own social security and social services. Although a systemic change towards a sustainable economy is not embedded in any of these models, the first type self-evidently promotes the growth of ESIs, which is visible in our research in several countries, too. This perspective of welfare policies leads to a brief look back at our original starting point in social work research, which searched for solutions in ESIs to the needs of the young generations at the margins of the

mainstream labour market. The sustainability impact of ESIs is not limited to only the short-term options of bringing people into formalised job-like activities in recycling projects. Furthermore, ESIs may open perspectives for a cultural value change since the meaning of work and alternatives to the destructive impacts of the mainstream economy on human wellbeing and environment may be reflected during the participation in ESIs [58]. If social work is committed to fighting poverty in an ecologically sustainable way, it should rather focus on supporting ESIs and similar community activities as relevant perspectives not only for the young people in unemployment. The major societal issues of income, recognition and lifestyles beyond full employment are so far interconnected with the search for sustainability transition in social work [59].

**Author Contributions:** Conceptualization, A.-L.M., I.S., T.H. and K.N.; methodology, A.-L.M., I.S., T.H. and K.N.; formal analysis, A.-L.M., I.S. and T.H.; investigation, A.-L.M., I.S., T.H. and K.N.; writing—original draft preparation, A.-L.M; writing—review and editing, A.-L.M., I.S., T.H. and K.N.; supervision, A.-L.M. and K.N.; project administration, A.-L.M.; funding acquisition, A.-L.M.

**Funding:** This research was funded by The Academy of Finland, grant number 285868.

**Acknowledgments:** We wish to acknowledge the Academy of Finland for funding the research project. We also appreciate our colleagues in the networks and the participants in the ESIs who provided us with access to the field and data. We also wish to acknowledge Nicholas Kirkwood for helping us proofread the text.

**Conflicts of Interest:** The authors declare no conflict of interest.

## Appendix A. List of Ecosocial Innovations (ESIs) by Country Included in the Mapped Data

Finland:

1. Valtaajat-projekti, Tatsi ry. Helsinki, http://valtaajat.fi/(TheSquatters)
2. Oma Maa, Organic food cooperative, Helsinki region, https://www.omamaa.fi/in-english/
3. Oma Pelto, Urban Co-operative FarmFood, Helsinki, https://www.omapelto.fi/english
4. Lapinlahden Lähde, cultural center, Helsinki, http://lapinlahdenlahde.fi/fi/main-page-3-2/
5. Kokkotyösäätiö, Foundation for work rehabilitation, Kokkola region, https://www.kokkotyo.fi/
6. Jyväskylän Uusiotuote, Recycling Workshop, Jyväskylä, http://www.uusiotuote.fi/
7. Lentoon/Take-Off -project, multi-placed program of training and employment (page no more available)
8. Töitä nuorille -kampanja/Jobs for the youngsters—campaign, Helsinki https://www.kierratyskeskus.fi/tietoa_meista/toihin_kierratyskeskukseen/toita_nuorille_-kampanja
9. Kestävän kehityksen keskus, Centre of Sustainable Development—Rehabilitation Center, Oulu, http://kestavankehityksenkeskus.net/järjestöt
10. Kulttuuripaja Elvis, Cultural rehabilitation workshop, Helsinki, https://niemikoti.fi/yksikko/kulttuuripaja-elvis/
11. Mun Juttu—hanke, youth participation -project, Lahti, http://www.lahdenyliopistokampus.fi/?s=Mun+juttu
12. VAMOS—youth project, multi-placed eight cities, https://www.hdl.fi/en/
13. Valoa elämään –hanke, training in work—project, 6 cities, www.valo-valmennus.fi
14. Kotkan pyöräpaja, bike workshop, Kotka, https://facebook.com/kotkanpyorapaja
15. Luontopolkua eteenpäin –työpaja, Nature path—workshop, Tampere, https://trety.org/luontopolku/
16. Porin luontopaja, Nature workshop, Pori, https://www.facebook.com/PorinLuontopaja/
17. Luontoa elämään—Kemijärven osahanke, Nature-based wellbeing -project, Kemijärvi https://www.lapinamk.fi/fi/Yrityksille-ja-yhteisoille/Tutkimus-ja-kehitys/Hyvinvointipalveluiden-osaamisala/Luontoa-elamaan#
18. Turun kirjakahvila, Literature Café, Turku, http://www.kirjakahvila.org/
19. Kulttuurikahvila Laituri, Cultural Café, Joensuu, https://www.facebook.com/laiturijoensuu/

20. Jupiter-säätiö, Foundation for training and employment, Vaasa (page no more available)
21. After eight Pietarsaari, Music and youth employment café, Pietarsaari http://aftereight.fi/ae/
22. Hirvitalo, Art and urban gardening, Tampere, http://www.hirvikatu10.net/wordpress.1/

Germany

23. KlimaWerkstatt Spandau, Climate workshop, Berlin http://www.klimawerkstatt-spandau.de/index.php
24. Kunst-Stoffe e.V., Up-cycling of materials for arts, Berlin, https://www.kunst-stoffe-berlin.de/
25. Life e.V., Ecological training, Berlin, http://www.life-online.de/
26. Foodsharing e.V., Germany, https://foodsharing.de/
27. Real Junk Food Project, Berlin, https://realjunkfoodberlin.wordpress.com/
28. Haus der Eigenarbeit (HEi), do-it-yourself-house, München, http://www.hei-muenchen.de/
29. Transition Berlin and Brandenburg, https://transitionberlinbrandenburg.wordpress.com/
30. Die Wille e.V., The Will—Work inclusion project, Berlin http://www.evangelisches-johannesstift.de/die-wille
31. Projekthaus Potsdam, House of projects, Potsdam, http://www.projekthaus-potsdam.de/index.php

UK

32. Local Projects Durham—Ideas for Change—Ideashive-page (page no more available)
33. Abundant Earth, food cooperative, Durham http://www.abundantearth.coop/
34. Empty Shop—cultural community meeting point, https://www.facebook.com/emptyshopHQ/
35. Re-f-Use, social café and enterprise against food waste, Durham, https://refusedurham.org.uk/
36. Incredible Edible Durham, permacultural gardening, https://www.incredibleedible.org.uk/find-a-group/durham-city/
37. Recyke Y'Bike, sharing economy with bikes, http://recyke-y-bike.org/
38. Transition Durham, https://transitiondurham.org.uk/

Belgium

39. De Winning, multi-located employment and agricultural project, Flanders, http://dewinning.be/
40. Velo, employment, re-cycling and mobility, Leuven; http://www.kuleuven.be/velo/_eng/
41. Arbeitscentrum De Wroeter, employment and agriculture, Flanders; http://www.arbeidscentrum-dewroeter.be/
42. De RuimteVaart, social restaurant, Leuven, http://www.deruimtevaart.be/
43. Social Innovation Factory, https://www.socialeinnovatiefabriek.be/nl/english

Italy

44. Akrat, up-cycling cooperative, Bolzano, https://akrat.squarespace.com/home/
45. CLAB, Bolzano, up-cycling cooperative and social enterprise, Bolzano, http://www.clab.bz.it/
46. Albatros, larger social cooperative, Merano, http://www.albatros.bz.it/de/index
47. WiaNui, small up-cycling enterprise, Brixen, http://www.wianui.eu/
48. Vinterra, organic agriculture and social cooperative, Mals, http://www.vinterra.it/
49. Bürgergenossensschaft, Obervinschgau, citizens' cooperative, Mals, http://www.bgo.bz.it/
50. Hollawint, Mals, organic agriculture and community gardening, http://hollawint.com.dedi4234.your-server.de/

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
