# Peer review of "Ecosocial Innovations and Their Capacity to Integrate Ecological, Economic and Social Sustainability Transition"

_sustainability, doi:10.3390/su11072107_

Round 1
Reviewer 1 Report
Dear authors,
This paper presents the contribution of ESI initiatives to suitability transformation in the context of social work research. I recommend for publication but with mayor revision on the text.
Introduction
1. Line 47. When the authors express aspects or areas of sustainability, they mention in the abstract: social, ecological and economic aspects (see Line: 19) while in Line 47 they introduce technical, I recommend considering the same sustainability aspects through the document.
2. Line 52. Their integrative capacity refers to what?
3. Line: 72. Maybe it is necessary to specify what these sustainability transition fields are
4. Line 109. It is not clear how does mentioning the gap of the socio-spatial relations contributes, if this is not discussed later.
5. Line 124. It is not clear what the authors mean by the comprehensive dynamic of the transformative capacity. Please explain more or rephrase the sentence.
6. Line 142. Add a reference.
Material and methods
7. Paragraph Lines 160-167. Are these the criteria used to select ESI? put it explicitly. Did the selected ESIs meet all the criteria? Or how did you evaluate it?
8. Lines 178-180. Is this paragraph the general objective of the paper? This could be integrated into paragraph Lines 72-79 to make the main objective clearer and stronger
9. Appendix 1 is missing
10. Lines 189-206. This paragraph seems more results than methods
11. In the methodology it is not clear how the second objective was addressed
Results
12. Line 252. Participants referred to what in this sentence?
13. It would be interesting to mention which of these 4 typologies of ESI is the most/least common, or which combinations of typologies are the most common.
14. Line 259. It is possible to quantify what part of the ESI had an ecological focus?
15. Line272. Rephrase the sentence, the sense is not clear
16. Line 327. Please be consistent with the quotation format through the document.
17. Line 354. A merger of what? Explain better.
18. Lines 390-409. The fifth factor is not clear. Is it a conceptual work on sustainability as it appears in Line 317? Or that explicitly addresses sustainable development (Line 390). Are these ideas the same? Then, on Line 396 the quotation is about innovation not about sustainability. Also in Line 403 conceptual activities are mentioned. Please review this paragraph so that the idea becomes more clear and consistent
Discussion
19. Line 412. Innovative potentials? Or innovative experiences?
20. Lines 418- 420. Add a reference
21. Line 430. The results did not mention gender, housing and energy issues that are mentioned in the discussion. I suggest mentioning them in results.
22. Line 430. Delete “l” before the word step
23. Line 436. There are two final dots..
24. Line 433. Climate change: okay, but ozone layer depletion: the relation with the practices mentioned it is not clear.
25. Line 436-446. These phrases seem disconnected. Reorder the ideas regarding the contribution of ESI initiatives to planetary (or ecological) sustainability and human wellbeing. Mention their contributions and their limitations.
Conclusions.
26. Line 480. Landscape changes? what does it mean?
27. Lines 476-484. This paragraph seems more a discussion of results than a conclusion.
As I am not a native English speaker, I can not give feedback on the English.
I hope my comments would contribute to the improvement of your paper,
I look forward to see the revised version,

Author Response
Dear Editors and Reviewers,
We highly appreciate the opportunity to re-submit a revised version of our paper Ecosocial innovations and their capacity to integrate ecological, economic and social sustainability transition
We found the insightful comments from both reviewers to be extremely helpful and relevant, and have got a feeling that after applying them, the paper has become much better.
As suggested by the reviewers, we have improved:
the research design by rewriting the introduction and making the research design more focused on the current research results on social innovations in sustainability transition research
the methods by adding more detailed description of the criteria for selecting the ESIs, the respective countries as well as the analysis and interpretation. We also added the Appendix, i.e. a list of the 50 ESIs in the same text file. It seems to us that unfortunately the reviewers did not receive the appendix which was submitted separately
the results by re-writing the discussion of the results and contextualising them with the previous research and also by stating the limitations of our data and analysis
the conclusions by rewriting them and focusing more clearly on policy suggestions, reflecting the research needs, and by re-connecting to the initial social work context.
The language of the re-written manuscript has been proofread by a native speaker specialist.
The comments from reviewer 1
All the detailed comments 1 – 27 have been applied to the manuscript and are visible as ‘tracked changes’, and we have also added comment box where the changes are. However, since a lot of text has been moved, re-moved or added, all our additional comments may not be visible any more.
We improved the sentences which were unclear and corrected the typing mistakes and applied to the following comments:
7. We added the detailed criteria used in the search for ESIs
8. We improved the objective of the paper and moved the phase from the Methods to the Introduction
9. We added the Appendix into the text file at the end
10. We moved the paragraph to the Results
13 – 14. We are grateful for the suggestion and added a quantification to the typology
18. We clarified the meaning of the fifth factor and used a data quotation which better describes it
27. We re-wrote the Discussion and Conclusion
The comments from Reviewer 2
we rewrote the Introduction according to the suggestions
we developed the methods a lot and added details about the ESIs and Country selection. We also discuss openly the limitations of the data
we re-wrote the Discussion a lot
we re-conceptualised the conclusions
However, although very relevantly suggested by Reviewer 2 – and we are very grateful for the two references provided – we did not deepen very much the discussion on the financial instruments of the ESIs and the circular economy in this paper. This was for two reasons. Firstly, the mapping data used in this paper does not include enough systematic information about the financial base of the 50 ESIs. This was asked for in all the ESIs, which we visited personally, but it was not available from all. We have only a very general picture about the financial side of the ESIs at this stage. Secondly, since in the meantime we have run six more detailed ESI case studies in all the countries, two separate articles regarding the economic base are currently in the writing process: The first is about the role of the social security and labour market instruments, as compared at the country level, and the second applies the theoretical models of new forms of sustainable economy to analyse our data. These articles are as of present forthcoming. The sent reference are helpful for these further papers, too.
With best regards,
The Authors

Reviewer 2 Report
Dear authors,
I think that there is a great work behind the presentation of this paper which investigates a very relevant topics (the role of ESIs towards ST). However, I suggest to carefully reconsider your research approach and resubmit the article after improving the following issues:
In the introduction, you need to connect the state of the art to your paper goals. Please follow the literature review by a clear and concise state of the art analysis. This should clearly show the knowledge gaps identified and link them to your paper goals. Please reason both the novelty and the relevance of your paper goals.Please eliminate those multiple references. After that please check the manuscript thoroughly and eliminate ALL the lumps in the manuscript. This should be done by characterizing each reference individually. This can be done by mentioning 1 or 2 phrases per reference to show how it is different from the others and why it deserves mentioning.
The research methodology seems underdeveloped. What is the rationale for focusing on the selected countries? How do you exactly selected the ESIs for your study? Methods should be described in detail in order to make the reader able to appreciate the work and related robustness behind your study. Indeed, I think the research procedure could be much more clearly described by means of a diagram also highlighting its potential and limit.
Discussion is a very important section for your paper and I appreciated the efforts of the authors to frame results within relevant literature. However, I would like to see more discussions about
· financial instruments for ESIs (very relevant field within ST literature: see https://www.sciencedirect.com/science/article/pii/S0040162517306716?via%3Dihub; https://www.sciencedirect.com/science/article/pii/S0959652617321650),
· circular and sharing economy (see: https://www.sciencedirect.com/science/article/pii/S0921800916300325; https://www.sciencedirect.com/science/article/pii/S0261517718302358
Conclusions I suggest to authors to readapt after paper revision and propose policy directions for ST
I hope that these suggestions can be useful for improving your paper towards a new submission.
Author Response

(The authors gave the same response as above.)

Round 2
Reviewer 1 Report
Thanks to the authors for this new version of the manuscript. The paper has improved substantially and it is now clearer. I recommend for publication but with minor revision on the text.
1. Although I am not a native English speaker, I detect some errors and for this I suggest accepting the changes made in the document and making a new revision of the writing and the language
For example:
Line 25: Change to integrative for to integrate
Line 95: Change analyse for analyze
2. Lines: 86-88. Improve the sentence redaction.
One option (but not the only)
The aim of our article is to contribute to the understanding of the complex interconnectivity between ecological, economic and social sustainability transition by investigating how the social innovations expressed by grass-root initiatives integrate the sustainability …
3. Lines 117-118. Improve the sentence redaction.
Besides the ecosocial paradigm, the research on sustainability transition and the concept and the role of social innovations in transition build the theoretical premises of this article.
One option (but not the only)
The theoretical premises of this article are based on ecosocial paradigm and the role of social innovations in sustainability transition
4. Lines 527-529.
In this paragraph authors mention the economic and social challenges that ESIs must address. Could you add to this paragraph something related to ecological challenges?
Author Response
TO THE REVIEWER 1, 2. round,
Thank you a lot for the careful review and the comments!
We have made the changes in your comments 1 - .4 as you have suggested. Further we have maintained another double language checking by ourselves and the native expert.
We hope the article could now be published.
Best,
the authors
1. Although I am not a native English speaker, I detect some errors and for this I suggest accepting the changes made in the document and making a new revision of the writing and the language
For example:
Line 25: Change to integrative for to integrate
Line 95: Change analyse for analyze
2. Lines: 86-88. Improve the sentence redaction.
One option (but not the only)
The aim of our article is to contribute to the understanding of the complex interconnectivity between ecological, economic and social sustainability transition by investigating how the social innovations expressed by grass-root initiatives integrate the sustainability …
3. Lines 117-118. Improve the sentence redaction.
Besides the ecosocial paradigm, the research on sustainability transition and the concept and the role of social innovations in transition build the theoretical premises of this article.
One option (but not the only)
The theoretical premises of this article are based on ecosocial paradigm and the role of social innovations in sustainability transition
4. Lines 527-529.
In this paragraph authors mention the economic and social challenges that ESIs must address. Could you add to this paragraph something related to ecological challenges?

Reviewer 2 Report
Thanks for the revised version of the manuscript and for addressing the issues raised.
Author Response
TO THE REVIEWER2
Thank you a lot for the second review of our manuscript.
We have now maintained another double language checking by ourselves and the native expert.
We hope the article could now be published.
Best,
the authors
---
English language and style
( ) Extensive editing of English language and style required
(x) Moderate English changes required
( ) English language and style are fine/minor spell check required
( ) I don't feel qualified to judge about the English language and style
Yes | Can be improved | Must be improved | Not applicable | |
Does the introduction provide sufficient background and include all relevant references? | (x) | ( ) | ( ) | ( ) |
Is the research design appropriate? | (x) | ( ) | ( ) | ( ) |
Are the methods adequately described? | (x) | ( ) | ( ) | ( ) |
Are the results clearly presented? | (x) | ( ) | ( ) | ( ) |
Are the conclusions supported by the results? | (x) | ( ) | ( ) | ( ) |
Comments and Suggestions for Authors
Thanks for the revised version of the manuscript and for addressing the issues raised.
Submission Date
15 February 2019
Date of this review
21 Mar 2019 13:37:59